# Sample Reduction for Physiological Data Analysis Using Principal Component Analysis in Artificial Neural Network

**Cid Mathew Santiago Adolfo** [1,*,†,‡] **, Hassan Chizari** [1,‡] **, Thu Yein Win** [2,‡] **and Salah Al-Majeed** [3]

1   School of Computing & Engineering, University of Gloucestershire, Cheltenham GL50 2RH, UK; hchizari@glos.ac.uk
2   Department of Computer Science and Creative Technologies, University of West England, Bristol BS16 1QY, UK; thomas.win@uwe.ac.uk
3   School of Computer Science, University of Lincoln, Lincoln LN6 7TS, UK; salah.almajeed@gmail.com
*   Correspondence: cidmathew.adolfo1@gmail.com or s1910163@glos.ac.uk; Tel.: +44-(0)-7735-612887
†   Current Address: 4 Fairhaven Street, Cheltenham GL53 7PL, UK.
‡   These authors contributed equally to this work.

**Abstract:** With its potential, extensive data analysis is a vital part of biomedical applications and of medical practitioner interpretations, as data analysis ensures the integrity of multidimensional datasets and improves classification accuracy; however, with machine learning, the integrity of the sources is compromised when the acquired data pose a significant threat in diagnosing and analysing such information, such as by including noisy and biased samples in the multidimensional datasets. Removing noisy samples in dirty datasets is integral to and crucial in biomedical applications, such as the classification and prediction problems using artificial neural networks (ANNs) in the body's physiological signal analysis. In this study, we developed a methodology to identify and remove noisy data from a dataset before addressing the classification problem of an artificial neural network (ANN) by proposing the use of the principal component analysis–sample reduction process (PCA–SRP) to improve its performance as a data-cleaning agent. We first discuss the theoretical background to this data-cleansing methodology in the classification problem of an artificial neural network (ANN). Then, we discuss how the PCA is used in data-cleansing techniques through a sample reduction process (SRP) using various publicly available biomedical datasets with different samples and feature sizes. Lastly, the cleaned datasets were tested through the following: PCA–SRP in ANN accuracy comparison testing, sensitivity vs. specificity testing, receiver operating characteristic (ROC) curve testing, and accuracy vs. additional random sample testing. The results show a significant improvement in the classification of ANNs using the developed methodology and suggested a recommended range of selectivity (*Sc*) factors for typical cleaning and ANN applications. Our approach successfully cleaned the noisy biomedical multidimensional datasets and yielded up to an 8% increase in accuracy with the aid of the Python language.

**Keywords:** principal component analysis (PCA); artificial neural network (ANN); multidimensional dataset; dimension reduction process; sample reduction process (SRP); receiver operating characteristic (ROC) curve; selectivity (*Sc*); sensitivity; specificity





## 1. Introduction

It is clear that biomedical sensors will keep improving exponentially, as they become more accessible; more readily available on the market; more intelligent; smaller and more compact; and integrated into personal belongings, such as cellular phones, watches, and eyeglasses. It is an outstanding contribution to human health, entertainment, the military, security, sports, and leisure and in analysing a patient's physiological data and their interpretation. These sensors are an integral part of biomedical devices; however, this innovation has an ever-evolving trend coupled along with challenges to perform intelligently [1]. The integration of these sensors, subjected to different environments in

both wired and wireless implementations, adds random noise to the system. The sudden upswing of the noise floor yields a degradation in the signal-to-noise ratio (C/N) and low energy bit per noise reference (Eb/No) during data processing. The drastic increase in the probability of error (Pe) eventually decreases the accuracy in such a system [2]. This interference results in flawed calculation and decision making, especially for artificial neural networks (ANNs) in biomedical applications [3].

An artificial neural network (ANN) is a part of a computing system that simulates the ability of the human neuron to learn the complex characteristics of the environment, to recognise patterns, and to generalise the inter-relationships between the features, including multidimensional datasets. It is part of the umbrella of artificial intelligence (AI) and deep learning, and it solves problems that would be impossible or difficult to solve by human effort or statistical criteria. ANNs have self-learning skills, enhancing their efficiency as more effective and reliable data become available to them. Primarily, ANNs enable the complex inter-relationships between the features within a given dataset to be identified and have seen widespread adoption in different applications, including biomedical and signal processing, which is readily and publicly available by wearable sensors [4].

Since data are everywhere—such as on the Internet—neglecting the integrity of a source of information poses a significant threat for ANNs to misinterpret the data. For critical aspects of the medical field, the efficacy of a diagnosis is threatened if the analysis that aids medical experts in research has a weak source, especially if dealing with the multitude of datasets that result in the presence of noisy and biased data, which often occurs from seemingly reliable sources [5].

Low-powered sensors are a critical source of unreliable information. Sudden changes in conditions can introduce environmental noise, and there are many possible avenues for noise and interference to blend into any part of the system. A sensor's scalability is another problem, given that the sensors nowadays are low power and with limited computation. Placement or location is another significant constraint, and if they are placed too close to other devices, they may be affected by crosstalk interference [6].

To understand the data, first, one needs to understand their composition. Data comprise valid and dirty data. Accurate data contain information that is accurate, holds predictive power, and is generalised to the entire set. Dirty data contain information that is misleading, noisy, or erroneous data, such as pragmatic contexts and the semantic- and syntactic-biased errors shown in Figure 1 [7].

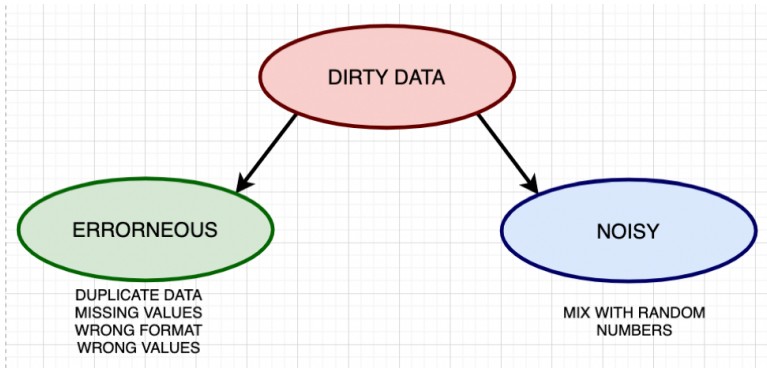

**Figure 1.** Dirty data.

Noisy data are any random fluctuations considered unwanted, unpleasant, loud, or disruptive that hinder the generalisation of the entire dataset. For an ANN with predictive power to analyse the data, the dataset needs to be free from noise to reduce errors in classification and prediction, which would significantly impact medical experts in their professional biomedical interpretation. Given recent technological solutions, one relevant control algorithm is highlighted in solving these challenges [8].

Principal component analysis (PCA) is usually part of data visualisation. PCA converts the features of datasets from high dimensional data and can help convert data to low-dimensional data with the aid of covariance, eigenvalues, and eigenvectors in identifying and sorting the strength of the predictive power of all features [9]. PCA allows for the extraction of critical feature vectors from multidimensional datasets in terms of data visualisation and loses some crucial information along the process [10], which is a considerable disadvantage, primarily if it is used extensively in the field of machine learning. This study used PCA not as a dimension reduction but as a sample reduction in order to remove the unwanted noisy samples in multidimensional datasets.

This paper proposes an application of a PCA–sample reduction process (SRP) to improve the prediction accuracy of ANNs. Using publicly available biomedical datasets from a different field provides qualitative analysis and demonstrates the effectiveness of said method in improving the accuracy of ANNs. By cleaning the publicly available biomedical datasets before training and testing the ANN, the classification problem is expected to increase the accuracy and to lower the computational cost, which is a great help in dealing with the the analysis and prediction of big multidimensional datasets.

A study to reinforce the methodology of an ANN using PCA–sample reduction is proposed in this paper to determine the proposed system's accuracy and performance with different sizes of multidimensional biomedical datasets. Furthermore, we investigate heart disease, voice and speech analysis for gender recognition, breast cancer classification, and cancer patient datasets to prove the versatility and flexibility of the proposed data-cleansing technique.

The rest of the paper is structured as follows. Section 2 provides a literature review of data cleaning in biomedical applications. Section 3 discusses the basic concepts of principal component analysis and related topics. Section 4 discusses how the PCA–SRP is integrated with an ANN. Section 5 provides the recommended *Sc* ranges. Section 6 interprets the result of publicly available biomedical datasets. Lastly, Section 7 provides a discussion, the conclusion, and future research directions.

## 2. Data-Cleaning Applications

The volume of data collected nowadays is vastly increasing, and since most data acquired are polluted, the dependability of the data is declining. Various data-cleaning methodologies are available to rectify this issue, but data cleansing remains difficult when working with large data requirements. Data cleaning, also known as data cleansing, is no longer a recent area of research. It aims to increase data quality by detecting and eliminating errors and inconsistencies [11]. As of now, there are two classifications of data cleansing: traditional data cleansing and data cleansing for big data. Traditional data cleansing techniques are so called because it is not used to manage massive volumes of data, such as Potter's Wheel and Intelliclean [12].

Meanwhile, the techniques in Table 1 such as Cleanix [13], SCARE [14], KATARA [15], and BigDansing [16] are developed specifically for big data. Regarding the emerging trends in data-cleaning techniques, one of the new challenges that researchers are about to face is scalability [17]. One of the perennial problems in data analytics is identifying and restoring dirty data, and failure to do so will result in faulty analytics and unreliable decisions. New abstractions and scalability are among the various facets of this issue and are considered when developing data-cleaning methods to cope with the amount and diversity of data [13,14,16]—see Table 2. Given the significant amount of data, it needs time to be processed to be suitable for big data analysis and decision making. The data's volume, veracity, and velocity must also be considered when analysing the proposed approaches; however, the researchers mentioned that "Data analytics is not about having the information known, but about discovering the predictive power behind the data collected" [12].

**Table 1.** Data-cleansing methods for big data.

| Methods | Key Features | Approach |
|---|---|---|
| Cleanix | Scalability, unification, and Usability | Rule Selection |
| SCARE | Scalability | Machine Learning Technique |
| KATARA | Easy Specification, Pattern Validation Data Annotation | Knowledge-base and Crowdsourcing |
| BigDansing | Efficiency, Scalability and ease of use | Rule Specification |

Cleanix, SCARE, and BigDansing focus on the scalability issue in the data cleansing process. Moreover, SCARE and BigDansing do not require any human-domain expert in the cleansing process. SCARE needs an extensive set of rules to update the dataset; however, no expert is present in the process. Nevertheless, the process is expensive, and if the authorities fail to identify correct fixes for the dirty dataset, it will result in redundancy of the training data and a threshold machine learning parameter that is hard to set precisely [15]. Furthermore, BigDansing also requires a set of data-quality rules for optimisation of the cleansing process that requires too many regulations to calibrate and put into place before the start of the cleaning process, as shown in Table 2 below; however, it needs no human-domain expertise to monitor the whole process, although adjusting such parameters is crucial in maintaining the essence of the information in the datasets [12].

**Table 2.** Currently used data-cleaning technique.

| Cleaning Technique Data | Volume Scalability | Veracity No need for Extensive Data Quality Rule Optimisation | Velocity No need for Human Domain Expert |
|---|---|---|---|
| Cleanix | x | | |
| SCARE | x | | x |
| KATARA | | x | x |
| BigDansing | x | | x |

These data-cleansing techniques support various data-cleaning tasks such as abnormal value detection and correction, incomplete data filling, de-duplication and conflict resolution (Cleanix), value modification (SCARED), the identification of correct and incorrect data, and the generation of top-k possible repairs for inaccurate data (KATARA) and rules into a series of transformations that enable distributed computations and several optimisations (BigDansing).

There are two kinds of dirty data: erroneous and noisy—see Figure 1. The current big-data-cleansing techniques—Cleanix [13], SCARE [14], KATARA [15], and BigDansing [16]—are focused mostly on erroneous dirty data such as a solution in duplicate entries, missing values, wrong values, and wrong formats. In medical applications, these big-data-cleansing techniques play a critical role in medical record management in medical facilities; however, they do not address noisy data acquired via biomedical sensors. Cleanix, KATARA, and BigDansing are not able to predict the correct values through a machine learning approach and cannot determine how to eliminate the mixed random number [12]; however, the SCARE technique, though constructed via machine learning, can only replace the missing values with the most precise value. When dealing with noisy data, the common approach in applications with machine learning and artificial neural network (ANN) is to address noise reduction and suppression [11,18,19].

Given all of the above data-cleaning techniques, noise and random sample reduction is a significant part of their scope of discussion, which is vital in biomedical data signal analyses acquired in wearable biomedical sensors due to its subjectiveness in a different kind of environment—an environment often filled with varying sorts of noises, such as thermal and acoustic noises and interference. Moreover, since most wearable biomedical sensors are of low power, by suppressing these kinds of noises through signal processing by applying the filtering threshold method, the unsupervised classification is not effective under a low SNR. When the spectral characteristics of the noise are so similar or near that of the sensor-received signal, the detection performance may be degraded [20]. When it applies to the classification problem of an artificial neural network (ANN), obtaining the correct values through comprehensive and extensive quantisation in data-signal processing is essential. Still, it is not sufficient to identify which of the gathered data have a predictive power, as some authors mention that "Data analytics is not about having the information known, but about discovering the predictive power behind the data collected" [12]. Correct values of the data do not guarantee that it holds a valid value in predicting and summarising the entirety of the multidimensional datasets. Mining those correct values is an integral part of data mining in every machine learning and big data analysis.

Complicated and straightforward mistakes are unavoidably present in data input and acquisition. Although much effort could be expended into this front-end procedure to reduce entry mistakes, the truth remains that mistakes in massive datasets are prevalent. Field error rates usually are about 5% or higher unless an organisation takes extraordinary precautions to prevent data inaccuracies [21]. Moreover, this rate is still quite high, at such a rate that it might lead to erroneous interpretation and decision making.

In the case of cleaning the noisy data via biomedical sensors, most researchers use principal component analysis, such as a reduction in dimensionality or feature space [22–26], feature extraction in further data visualisation [27–29], and feature selection tools in machine and deep learning applications [30–34] for machine learning applications.

Principal component analysis (PCA) is usually utilized in dimension or feature reduction and provides a significant increase in accuracy and efficiency along with other machine learning techniques in many applications aside from biomedical application [35–39]. Nevertheless, some of the information is lost during the process of dimension reduction [10].

Figure 2 shows a sample Excel dataset, and PCA reduces the dimensions vertically (by column), shown in red; for the proposed methodology, it reduces the noisy samples horizontally (by rows). We emphasise that the extraction of features, such as covariance, eigenvalues, eigenvectors, and dimension reduction, is not a novel technique that we propose here, but instead, we propose the implementation of sample reduction using PCA. Identifying the noisy samples that cause irregularity in the multidimensional datasets and omitting or reducing them is the main focus of this study.

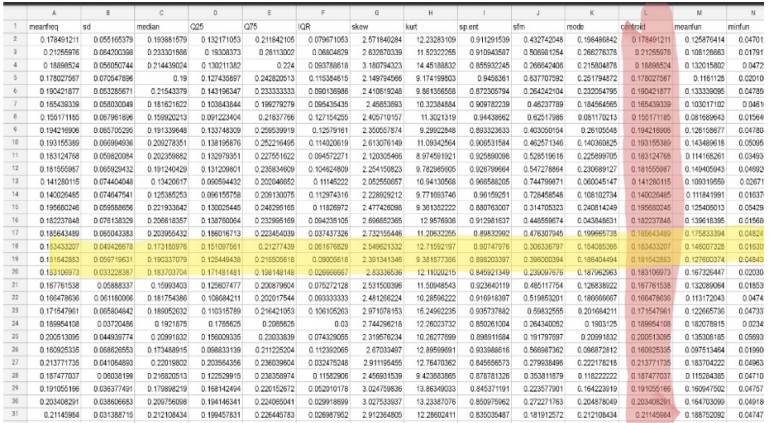

**Figure 2.** Sample Excel multidimensional dataset.

The observation and the application of the PCA–sample reduction process are utilised for data cleansing of the noisy multidimensional datasets to increase the accuracy of classification problems in artificial neural networks and to identify recommended threshold ranges. To the best of our knowledge, we are the first to propose this unique approach. We discuss a specific technique for qualitative noise detection and omission of such. These techniques are explained with a motivating example highlighting deep learning classification problems of artificial neural networks in biomedical applications using publicly available biomedical datasets. The simplicity of this technique makes it portable and can apply to a variety of tasks for fast and accurate classification of typical or commonly used artificial neural networks.

In terms of software implementation, developers and data engineers prefer Python programming. It is the best option for projects or programming involving AI and big data analysis as Python is a simple language with a mature and supportive Python community; an abundance of support from renowned corporate sponsors; an extensive and popular selection of libraries; and the ability to work with heavy-hitting frameworks such as TensorFlow, Sci-kit-learn, OpenCV, and Keras [40].

## 3. Principal Component Analysis–Sample Reduction Process

Principle component analysis (PCA) aims to minimise the dimensionality of a dataset with many connected variables while keeping as much variance as feasible. The conversion of the new collection of uncorrelated variables known as principal components (PCs) preserves most of the variance included in the original variables. A new set of dimensions or orthogonal measurements are linearly independent and ranked according to the data variance. This means the more crucial principal axis occurs first (more important = more variance/more spread-out data). In general, understanding the PCA, variance, covariance, eigenvalues, and eigenvectors has an essential role in this concept [41].

Figure 3 shows large positive covariance means that X and Y are completely related, i.e., as X increases, Y also increases. Negative covariance portrays the exact opposite relation. Zero covariance means that X and Y have no relation.

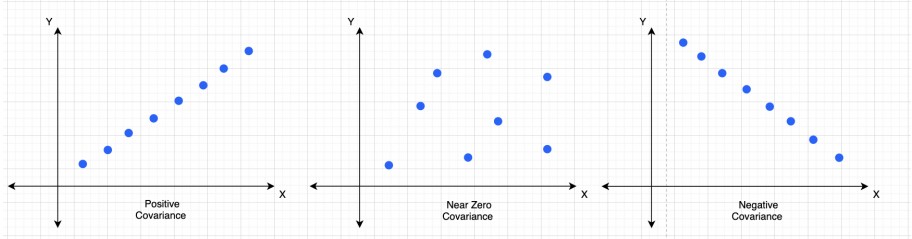

**Figure 3.** Covariance and their values.

The visualisation of data is an excellent approach to understand the patterns that lie in the multidimensional datasets. When information is placed in the horizontal and vertical axes (two-dimensional plane), it is straightforward to understand and discern the pattern that lies in it; however, the difficulty of conceiving it visually in multidimensional data with many features to consider and computing the data analysis becomes complex. Principal component analysis prerequisites require the discovery of patterns between the datasets so that the data are distributed across each dimension by first analysing the contributions of each feature in providing information to the overall dataset through eigenvector analysis. It then performs dimension reduction by keeping the feature columns with the highest eigenvalues.

To demonstrate the efficacy of PCA, Figure 4, which projects a set of multidimensional data onto a two-dimensional space, is used as an example. Due to the high dimensional nature of the data points, it can be challenging to identify a linear correlation between the data points. The points are represented as column vectors before being aggregated into a matrix $M$. PCA is then applied to $M$ as shown in Equation (1):

$$PCA(M) = W \times V \tag{1}$$

where $W$ and $V$ are the corresponding eigenvalues and eigenvectors, respectively.

$V_{PC_1}$ is generated using covariant eigenvalues $V$ with the corresponding top $k$ eigenvalue in $W$, which best represents the data points, and are then selected, before plotting them as shown in Figure 5 [30]. The principal axes ($PC_1$ and $PC_2$) are denoted by the red lines passing through the points that provide a graphical representation of the covariance amongst the points.

Lastly, the vital part of the proposed PCA–sample reduction process ($PCA_{SRP}$) is as follows:

$$PCA_{SRP}(M) = |M \times V_{PC_1}| \tag{2}$$

where $PCA_{SRP}(M)$ is the loading score of each sample in dataset $M$ and $V_{PC_1}$ is the $PC_1$ eigenvectors of the samples in matrix $M$.

$$S = \sum_0^D \begin{cases} 1 & \text{if } PCA_{SRP}(M) > Sc \times PCA_{SRP}(M)_{highest} \\ 0 & \text{otherwise} \end{cases} \tag{3}$$

$S$ is the number of samples above the loading score of set-biased $Sc$, and $S \leq D$, where $D$ is the number of samples in the dataset, including the random and noise samples.

$$passrate = S/D \tag{4}$$

The *passrate* is the rate of samples cleaned using PCA–SRP in dataset $M$ based on Equation (4). Any samples above said threshold are accepted, and those below it will be rejected for the new process dataset $D_{PS}$ as per Equation (3).

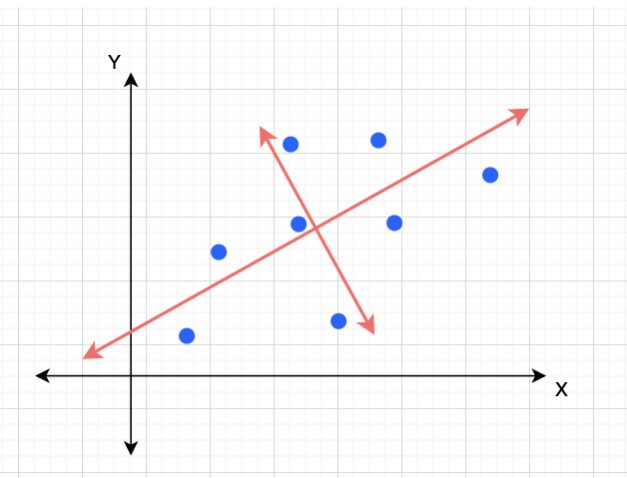

**Figure 4.** Projection of data points in two dimensions.

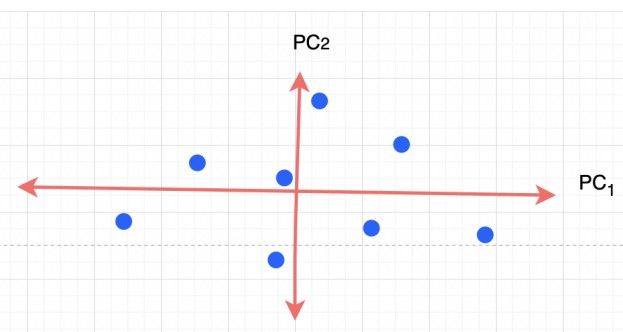

**Figure 5.** Projection of data points in the two newly constructed dimensions.

## 4. PCA–SRP Implementation in ANN

The proposed framework addresses the limitations of the ANN in processing multidimensional datasets through the use of PCA with the sample reduction process (PCA–SRP). PCA is used to analyse the multidimensional data, and if there exists a significant relationship between the features of a dataset, it is arranged from the most significant samples to the least to be visualised and the multidimensional data are put into perspective. It focuses on implementing dimension reduction by removing data from the multidimensional dataset feature columns that do not have high inter-feature covariance; however, the proposed framework using the loading scores can dissect all of the samples or rows in the multidimensional dataset. Figure 6 provides a graphical description of the proposed PCA–SRP-based ANN solution.

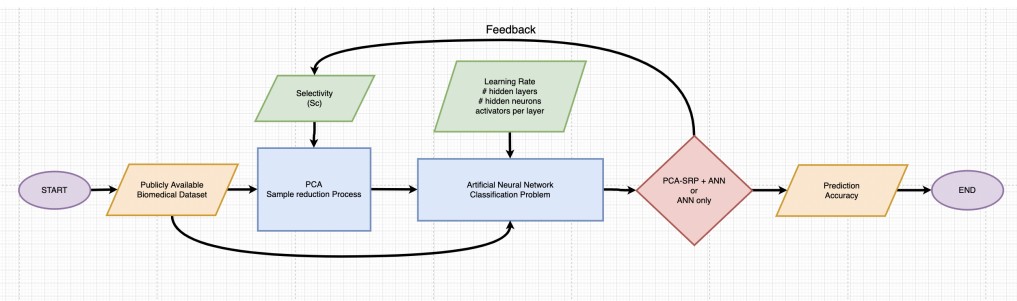

**Figure 6.** Conceptual flowchart of the proposed approach.

The PCA–SRP Python program generates a new set of multidimensional data with fewer samples based on the screen plot to identify the most significant samples. It is used in the Python ANN program as an input in order to ensure correct classification.

Figures 7 and 8 show the algorithm used in Python in the implementation and analysis of the concept. The implementation starts by extracting the data in .csv format and by injecting it into the PCA–SRP process. Major principal components are extracted through the threshold mentioned above based on the input dataset, resulting in a new set of multidimensional datasets used in ANN.

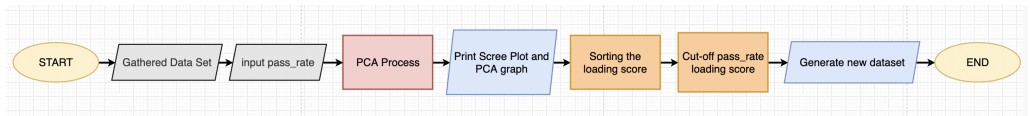

**Figure 7.** Detailed breakdown of the steps in PCA.

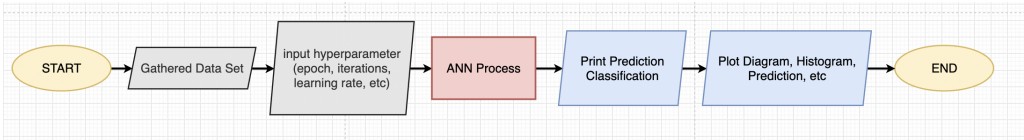

**Figure 8.** Detailed breakdown of the steps in ANN.

The proposed PCA–SRP utilised in the ANN consists of two hidden layers consisting of 32 neurons and 16 neurons. The learning rate was set at 0.1 and trained for 100 epochs. The input parameters in the PCA–SRP-based ANN are shown in Table 3 [42].

**Table 3.** PCA–SRP and ANN parameters.

| PCA–SRP and ANN Parameters | Values |
| --- | --- |
| Most significant sample size | $>Sc \times (PCA_{SRP}(M)_{highest})$ |
| Learning rate | 0.1% |
| Epochs | 100 |
| Number of ANN neurons | 2 hidden layers (32 and 16, respectively) |
| Activation functions used | ReLU in hidden layers SoftMax in final layer |

The accuracy of the PCA–SRP-based ANN was compared against a model trained using ANN alone on different datasets.

## 5. Sc Range Identification

Accuracy testing was conducted for the specified dataset in order to identify the $Sc$ range shown in Figure 9, to maximise the number of samples of the cleaned set $S$, and to maximise the number of samples of the removed set $R$ in the processed $D_{PCA-SRP}$.

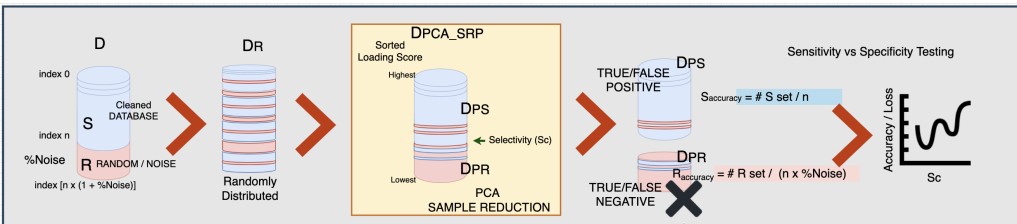

**Figure 9.** Detailed sensitivity vs. specificity testing procedure.

Assume that there are two sets of data $D$: cleaned data $S$ and $R$.

$$D = S + R \tag{5}$$

$$\%Noise = R/S \tag{6}$$

$$R = \%Noise \times S \tag{7}$$

Therefore,

$$D = S + \%Noise \times S \tag{8}$$

$$D = S \times (1 + \%Noise) \tag{9}$$

By indexing the dataset $D$ for identifier, where $n$ is the number of cleaned data $S$, we have the following:

$$D = \begin{cases} S & \text{if } 0 \leq \text{index} \leq n \\ R & \text{if } n < \text{index} \leq n \times (1 + \%Noise) \end{cases} \tag{10}$$

$D_R$ randomly distributes $R$ in $S$ to thoroughly mix the noise in the cleaned data by maintaining its index for identification. Then, $D_R$ is processed in PCA–sample reduction and sorted based on its loading score through the concept of covariance, eigenvalues and eigenvectors, and $D_{PCA-SRP}$.

$Sc$ is the selectivity of the data based on the equation in Table 3, $D_{PCA-SRP}$, where the cut-off is biased to separate the clean-processed data $D_{PS}$ and to remove processed data $D_{PR}$. The dataset was transformed from random to sorted PCA-SR-processed data.

Transformation:

$$D \rightarrow D_R \rightarrow D_{PCA-SRP} \tag{11}$$

Thus,

$$D = S \cup R \tag{12}$$

$$D_{PCA-SRP} = S_{PCA-SRP} \cup R_{PCA-SRP} \tag{13}$$

By definition, $S_{accuracy}$ (true positive rate or sensitivity) is the number of correct samples in a clean-processed $S$ in $D_{PS}$ with a relative to number of $n$.

$$S_{accuracy} = S_{D_{PS}}/n \tag{14}$$

However, $R_{accuracy}$ (True Negative Rate or Specificity) is the number of correct samples of $R$ in $D_{PR}$.

$$R_{accuracy} = R_{D_{PR}}/(n \times \%Noise) \tag{15}$$

Therefore, the most efficient $Sc$ is where $S_{accuracy}$ and $R_{accuracy}$ meet.

### 5.1. Test Selectivity ($Sc_{test}$)

To find $Sc_{test}$, the actual cleaned $S$ set and random $R$ set are needed to identify the accuracy by identifying the $S_{accuracy}$ and $R_{accuracy}$ lines and by applying Equation (16).

$$Sc_{test} = index[max(S_{accuracy} \times R_{accuracy})] \tag{16}$$

For $Sc_{min}$, $S_{accuracy}$ and $S_{accuracy}$ are maximised:

$$\frac{\partial(S_{accuracy})}{\partial(Sc)} = 0 \tag{17}$$

$$\frac{\partial(R_{accuracy})}{\partial(Sc)} = 0 \tag{18}$$

As shown in Equation (16),

$$F_{value} = S_{accuracy} \times R_{accuracy} \tag{19}$$

$$\frac{\partial F_{value}}{\partial(Sc)} = S_{accuracy} \times \frac{\partial(R_{accuracy})}{\partial(Sc)} + R_{accuracy} \times \frac{\partial(S_{accuracy})}{\partial(Sc)} = 0 \tag{20}$$

$$\frac{\partial^2 F_{value}}{\partial(Sc)^2} < 0 \tag{21}$$

Maximum $F_{value}$ with respect to $Sc$ is the $Sc_{test}$ by satisfying the Equation (21).

Note: Be cautious when using $Sc_{test}$ as $Sc$; it might remove all of the unwanted sample sets but may lose some information.

### 5.2. Minimum Selectivity ($Sc_{min}$)

The minimum $Sc$ is the rate of samples in $PC_1$, which shows that all included samples pass the minimum requirements of having the same attributes as the entire set.

$$Sc_{min} = PC_1 rate \tag{22}$$

Note: By using $Sc_{min}$ as $Sc$, while it might not lose important information, the dataset $D$ includes an abundance of random and noise samples.

The acceptable $Sc$ is shown in Equation (23) in Figure 10.

$$0 \geq Sc_{min} \geq Sc \geq Sc_{test} \geq 100 \tag{23}$$

Then,

$$Sc_{min} \geq Sc \geq Sc_{test} \geq 100 \tag{24}$$

Therefore, the recommended $Sc$ is between the range given in Equation (25). It varies depending on how much cleaning and removing are needed in the dataset $D$.

$$Sc_{min} \geq Sc \geq 100 \tag{25}$$

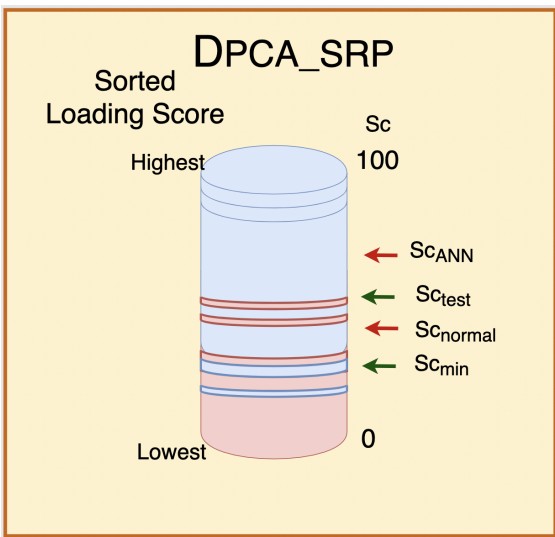

**Figure 10.** Sc—accuracy testing representation.

### 5.3. ANN Selectivity ($Sc_{ANN}$)

The ANN requires samples that allow for strong predictive power (see Section 7 for more details concerning the ANN requirements); we assumed the following:

$$Sc_{test} \geq Sc_{ANN} \geq 100 \tag{26}$$

However, $Sc_{test}$ is unknown due to unidentifiable $R$ samples, and it is suggested that $Sc_{ANN}$ is close to 100%.

### 6. Multidimensional Datasets

The multidimensional datasets for classification problems are tested under two layers of artificial neural network with 32 and 16 neurons, respectively, using ReLU and SoftMax activators, a testing size of 20%, a training size of 80%, 100 epochs, a learning rate of 10% [42], and $Sc$ = 98% of the highest loading score in PCA analysis. Noisy data based on a standard distribution are added to the original dataset before training to simulate noisy data samples. Figure 11 shows how a given multidimensional dataset is preprocessed and used in the proposed PCA–SRP approach. Each multidimensional dataset or corpora has selected the variation in its dimensions, primarily sample sizes; the number of classifications shown in Table 4; the noise sample contents; and the purpose it serves in any field of biomedical applications in classification problems.

**Table 4.** Datasets used and their metadata.

| Datasets | Number of Dimensions | Sample Size | Target Classification |
|---|---|---|---|
| Heart Disease | 14 | 301 | 2 |
| Gender Voice Recognition | 21 | 3167 | 2 |
| Breast Cancer | 31 | 568 | 2 |
| Cancer Patients | 24 | 1098 | 3 |

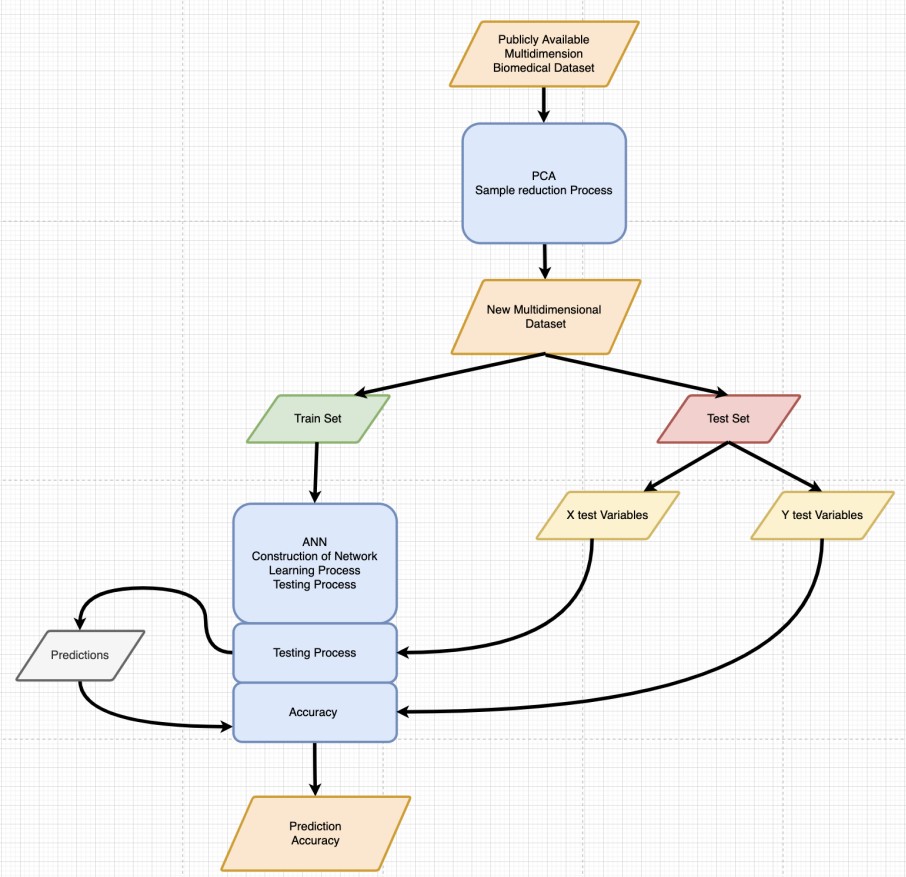

**Figure 11.** Pre-processing an application of multidimensional datasets in the proposed approach.

The different multidimensional datasets are shown in Table 4 and were acquired from different scientific and medical laboratories [43–47]. Specifically, to test our procedure, we vary the number of samples and dimensions. Moreover, researchers also are trying to show that the Python implementation works with a variety of dataset sizes.

### 6.1. Heart Disease Dataset

The experiment with the Cleveland patients dataset is concentrated on simply attempting to distinguish presence (values 1) from absence (values 0) to find other trends in heart data to predict certain cardiovascular events or to find clear indications of heart health [43].

Figures 12 and 13 show the result of the scree plot based on the values, indicating the PCA value distributions of the features in the dataset and the *Sc*-threshold cutoff limit, respectively. Based on the values, 78.48% of the top samples were selected in the training of the ANN model for a 98% selectivity.

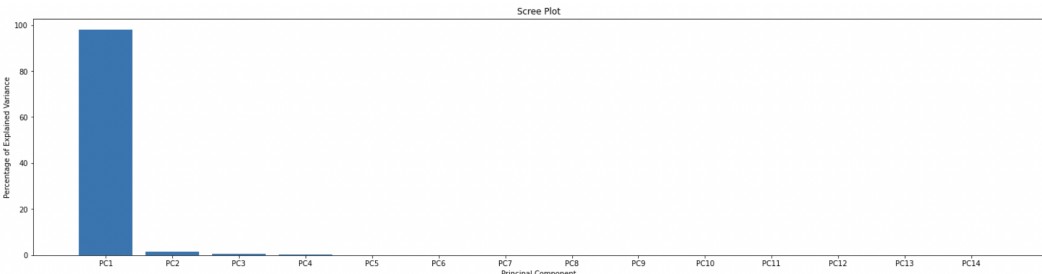

**Figure 12.** Scree plot of the heart disease dataset.

```
the sorted samples based of loading scores
 278      0.057987
226      0.057981
2        0.057976
45       0.057968
42       0.057961
             ...
147      0.003164
144      0.001178
138      0.000718
139      0.000605
327      0.000470
Length: 330, dtype: float64
```

**Figure 13.** Threshold cutoff limit of the heart disease dataset.

*6.2. Gender Voice Recognition Dataset*

The gender voice recognition dataset is based upon acoustic properties of the voice and speech to identify a voice as male or female. The dataset consists of 3167 recorded voice samples collected from male and female speakers. The voice samples are preprocessed by acoustic analysis in R using the Seewave and TuneR packages, with an analysed frequency range of 0–280 hz (human vocal range) [44].

Figures 14 and 15 show the threshold cutoff limit and the result of the scree plot show the PCA distributions of the features in the dataset, respectively. In the training of the ANN model, the proposed method selected 23.82% of the top samples from the dataset.

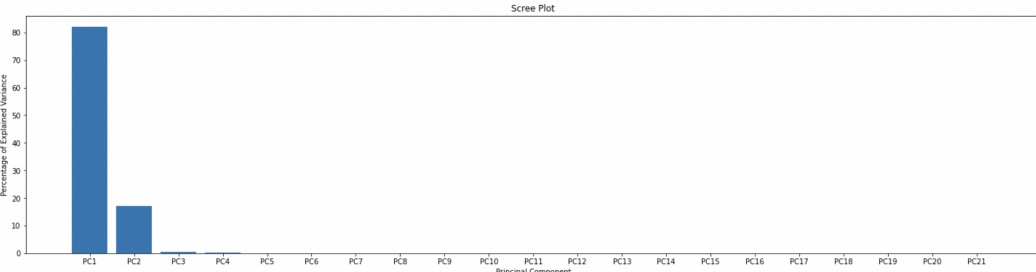

**Figure 14.** Scree plot of the gender voice recognition dataset.

```
the sorted samples based of loading scores
 2462      0.019580
3121      0.019578
2729      0.019575
2022      0.019574
2711      0.019571
             ...
1601      0.000244
1626      0.000243
3259      0.000181
1641      0.000168
3267      0.000068
Length: 3300, dtype: float64
```

**Figure 15.** Threshold cutoff limit of the gender voice recognition dataset.

### 6.3. Breast Cancer Classification Dataset

The computed dataset consists of a series of digitised images of a mass breast fine needle aspirate (FNA). They describe characteristics of the cell nuclei present in the three-dimensional image as described in [45].

Figures 16 and 17 show the threshold cutoff limit and the results of the scree plot showing the PCA distributions of the features in the dataset, respectively. The selection of the top eigenvalues in the samples obtains 93.67% of the original dataset.

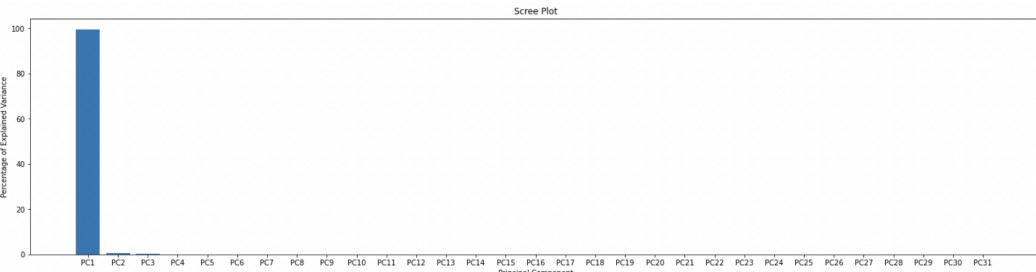

**Figure 16.** Scree plot of the breast cancer classification.

```
the sorted samples based of loading scores
 466     0.041997
545      0.041996
224      0.041996
438      0.041995
 93      0.041995
         ...
580      0.001718
578      0.001383
594      0.000928
576      0.000828
595      0.000477
Length: 600, dtype: float64
```

**Figure 17.** Threshold cutoff limit of the breast cancer classification dataset.

### 6.4. Cancer Patients Dataset

The data comprise information about hundreds of cancer patients and their lifestyles. It consists of three classes (low, medium, and high severity), based on the cancer patients dataset [47].

Figures 18 and 19 provide the threshold cutoff limit and a graphical description of the PCA distributions amongst the features in the dataset, respectively. The selection of high eigenvalues features resulted in 45.41% of the samples in the original dataset used for the model's training.

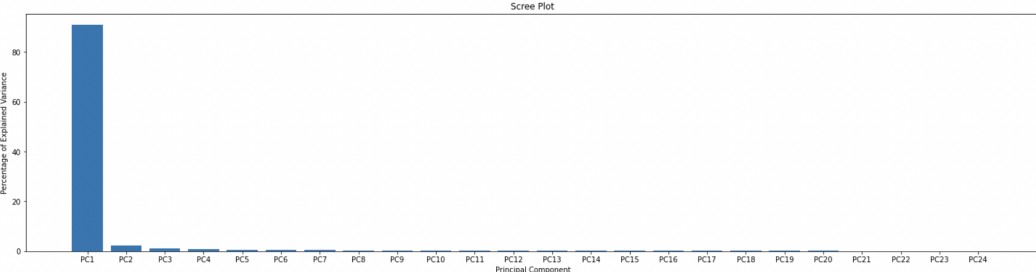

**Figure 18.** Scree plot of the cancer patients dataset.

```
the sorted samples based of loading scores
 900      0.031442
776       0.031442
332       0.031442
443       0.031442
110       0.031442
          ...
1082      0.006573
1061      0.006205
1043      0.004993
816       0.001869
98        0.000278
Length: 1099, dtype: float64
```

**Figure 19.** Threshold cutoff limit of the cancer patients dataset.

## 7. Discussion and Results

Upon acquiring the given datasets, they are continimated with noise and random samples. By definition, noise is small, unwanted forms of energy or samples; on the other hand, random samples are the unconscious and unspecified values within the range of normal values.

The datasets with noise and random samples are processed using PCA–SRP with 98% selectivity ($Sc$), shown in Table 5, and then subjected to following tests:

- PCA–SRP + ANN comparison accuracy testing compares the validation model accuracy with and without PCA–SRP in an ANN.
- Sensitivity vs. specificity testing is a diagnostic test to find the approximate $Sc$ range values ($Sc_{test}$).
- Receiver operating characteristic (ROC) curve testing compares the methodology PCA–SRP in different datasets in terms of organisation and classification of samples. Moreover, ROC curves also provide a practical evaluation of machine learning techniques.
- Accuracy vs. additional random samples testing is a diagnostic test responding to the sudden increase in noise and random sample spikes.

**Table 5.** Datasets sample status after PCA–sample reduction process.

| Dataset | Samples Used | %Passrate |
|---------|--------------|-----------|
| Heart Disease | 259/330 | 78.48 |
| Gender Voice Recognition | 786/3300 | 23.82 |
| Breast Cancer | 562/600 | 93.67 |
| Cancer Patients | 499/1099 | 45.41 |

### 7.1. PCA–SRP + ANN Comparison Accuracy Testing

We compared the validity model accuracy using PCA–SRP in an ANN classification problem as suggested by the results presented in Figure 20 and determined its effect upon being subjected to noise and random samples. Tables 6 and 7 display the validation accuracy using ANN + PCA–SRP and ANN only, showing a significant increase as shown in Figure 21 for both noise and random samples.

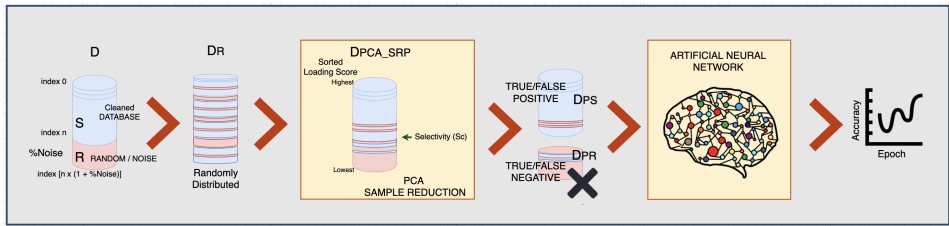

**Figure 20.** PCA–SRP to ANN process.

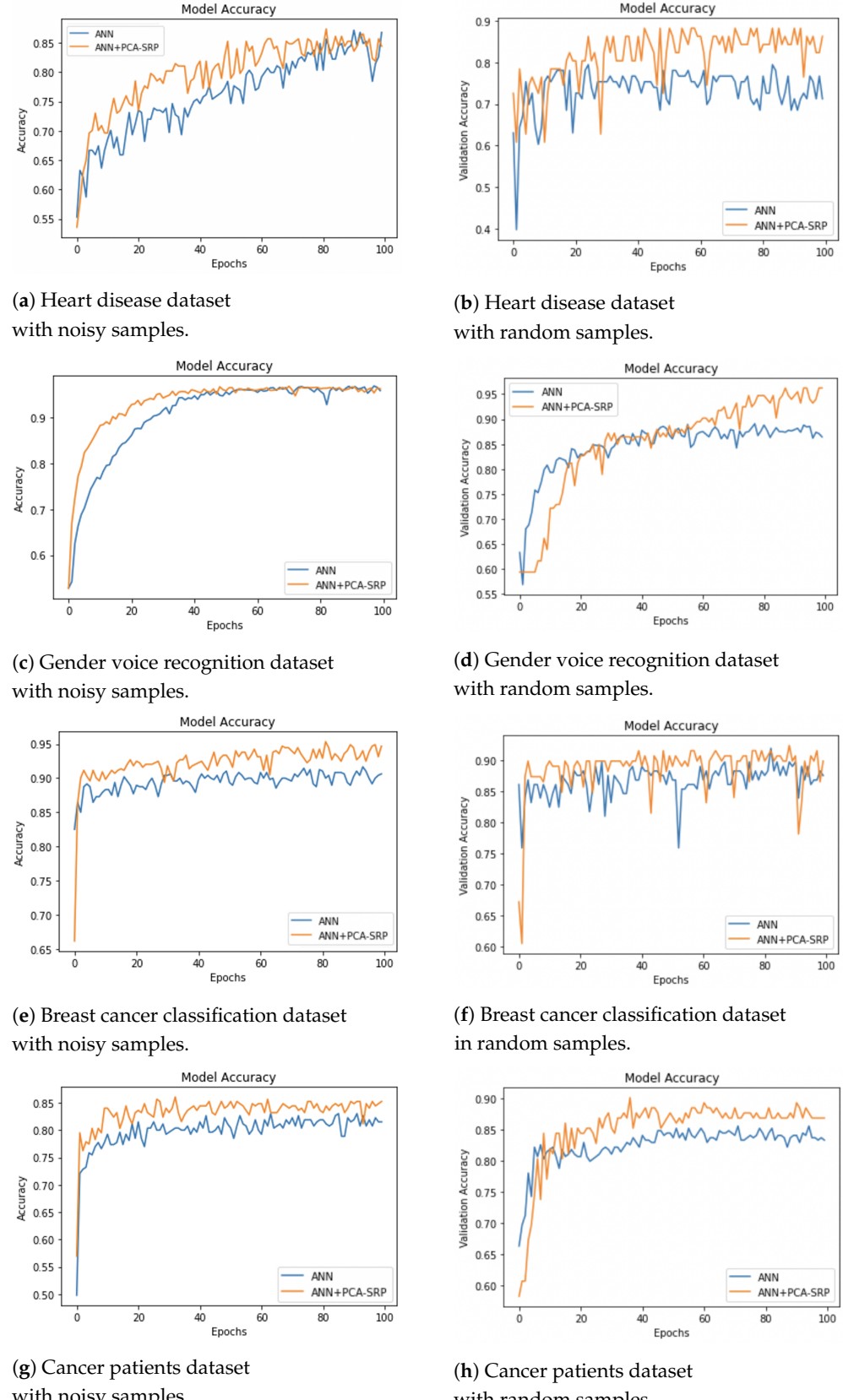

**Figure 21.** Validation model accuracy of the given datasets.

The difference between dataset accuracy subjected to noise and random samples are the definition of the accuracy curve line between ANN + PCA–SRP to ANN only,

even though both of them have a significant accuracy increase. The accuracy of datasets under the influence of random samples swings and deviates more and is less defined in comparison with accuracy with noise samples.

**Table 6.** A ccuracy of datasets with noisy samples.

| Dataset | ANN Only | ANN with SRP | % Increase |
|---|---|---|---|
| Heart Disease | 76.19 | 79.89 | 3.7 |
| Gender Voice Recognition | 90.6 | 93.29 | 2.69 |
| Breast Cancer | 90.49 | 92.36 | 1.86 |
| Cancer Patients | 79.82 | 83.27 | 3.45 |

**Table 7.** Accuracy of datasets with random samples.

| Dataset | ANN Only | ANN with SRP | % Increase |
|---|---|---|---|
| Heart Disease | 73.69 | 81.76 | 8.07 |
| Gender Voice Recognition | 85.72 | 91.2 | 5.42 |
| Breast Cancer | 83.3 | 86.46 | 3.12 |
| Cancer Patients | 80.90 | 88.81 | 7.91 |

### 7.2. Sensitivity vs. Specificity Testing

Sensitivity measures how many true positives remain in $S$ set, and it is described as a sudden dip as $Sc$ increases. Specificity measures how many true negatives remain in the removed $R$ set and increases as $Sc$ increases. Figure 22 presents the diagnostic testing yielding a recommended $Sc$ range in normal cleaning and ANN applications, as demonstrated in Equations (25) and (26), respectively.

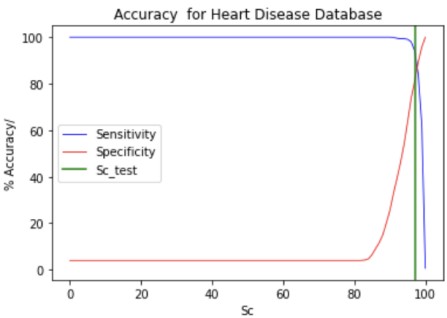

(**a**) Heart disease dataset.

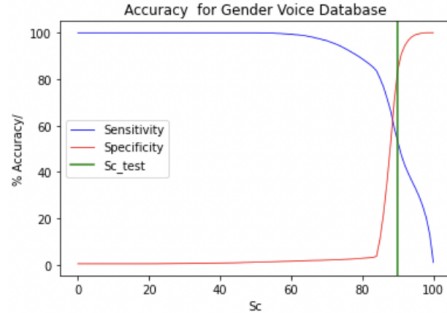

(**b**) Gender voice recognition dataset.

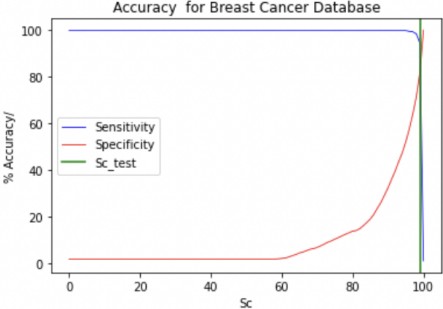

(**c**) Breast cancer classification dataset.

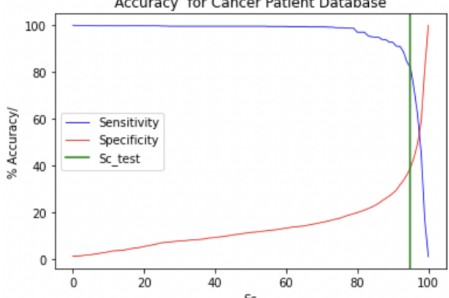

(**d**) Cancer patients dataset.

**Figure 22.** Sensitivity vs. specificity diagram.

Table 8 shows the relation between $Sc_{min}$ and $Sc_{test}$, but in practicality, $Sc_{test}$ is hard to determine due to unknown $R$ sets, and whether it consists of noise or random samples. In general, for the normal cleaning process, $Sc$ is close to or above the $Sc_{min}$ or $PC_1$ rate. For the given datasets, it is suggested that $Sc_{normal}$ is as follows:

$$80\% \leq Sc_{normal} \leq 96\% \tag{27}$$

**Table 8.** $S_{accuracy}$ ($Sc_{test-min}$) result.

| Dataset | $Sc_{min}$ | Sensitivity | Specificity | $Sc_{test}$ | Sensitivity | Specificity |
|---|---|---|---|---|---|---|
| Heart Disease | 96 | 98.04 | 73.68 | 97 | 94.15 | 81.04 |
| Gender Voice Recognition | 80 | 88.75 | 2.87 | 90 | 53.08 | 84.103 |
| Breast Cancer | 96 | 99.47 | 57.64 | 99 | 94.51 | 81.84 |
| Cancer Patients | 88 | 93.83 | 25.86 | 95 | 81.8 | 39.1 |

However, ANN applications need highly predictive samples [12], so it is suggested that selectivity ($Sc$) should be almost near 100%. Table 8 shows $Sc_{ANN}$ at the following range:

$$90\% \leq Sc_{ANN} \leq 99\% \tag{28}$$

A high $Sc$ value loses information and true positives in the $S$ set but increases the specificity of the dataset; however, a low value of $Sc$ adds noise and random samples that yield fewer true negatives and increases sensitivity. Careful adjustment of $Sc$ vouches for a good result as a cleaning agent in the system.

### 7.3. Receiver Operating Characteristic (ROC) Curve Testing

As observed in Figure 23, receiver operating characteristic (ROC) datasets have shown an organisation of the samples, even the random and noise samples were strongly mixed into the cleaned set. The larger the area under the curve (AUC), the better the classifier methodology for the true positive rate (sensitivity) vs. true negative rate ($1 -$ specificity) diagram, as seen in Figure 24. Ideally, the objective is the perfect classifier; nevertheless, a result above the random classifier line would allow us to conclude that the methodology is acceptable.

Given all of the datasets, the cancer patients dataset is acceptable but by not as much as the other datasets; it has a smaller AUC.

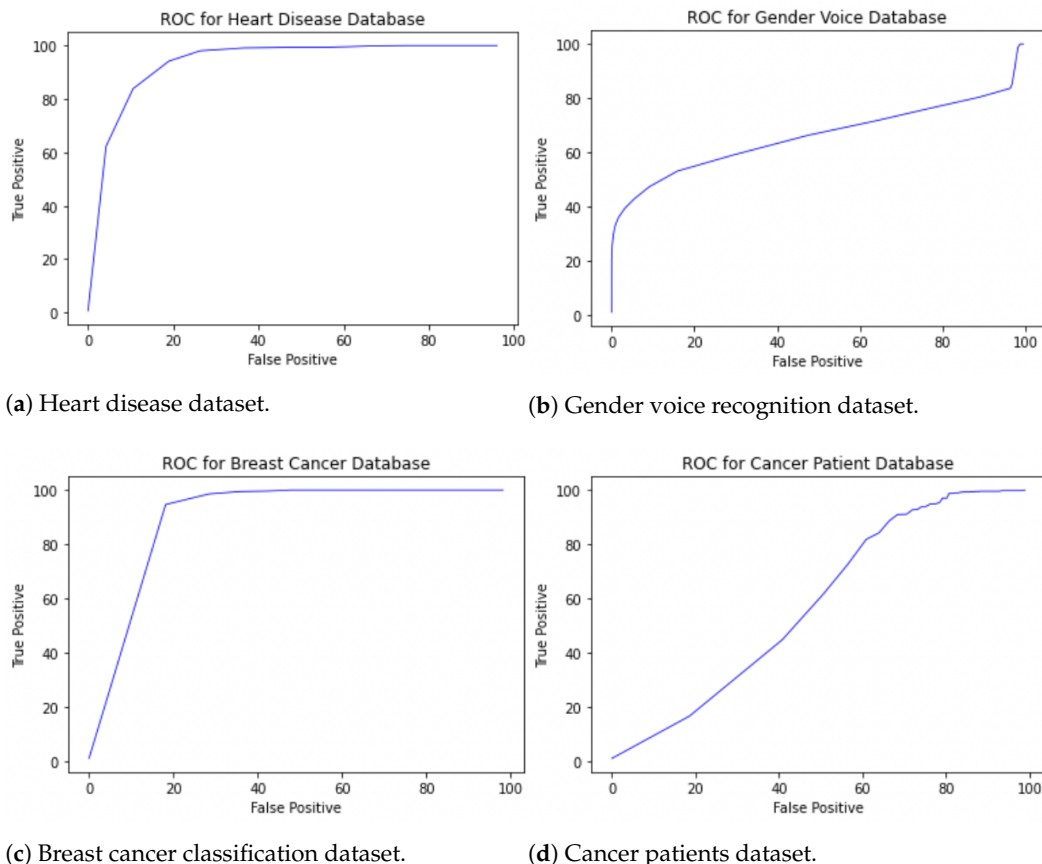

(**a**) Heart disease dataset.      (**b**) Gender voice recognition dataset.

(**c**) Breast cancer classification dataset.      (**d**) Cancer patients dataset.

**Figure 23.** Receiver operating characteristic (ROC) curve.

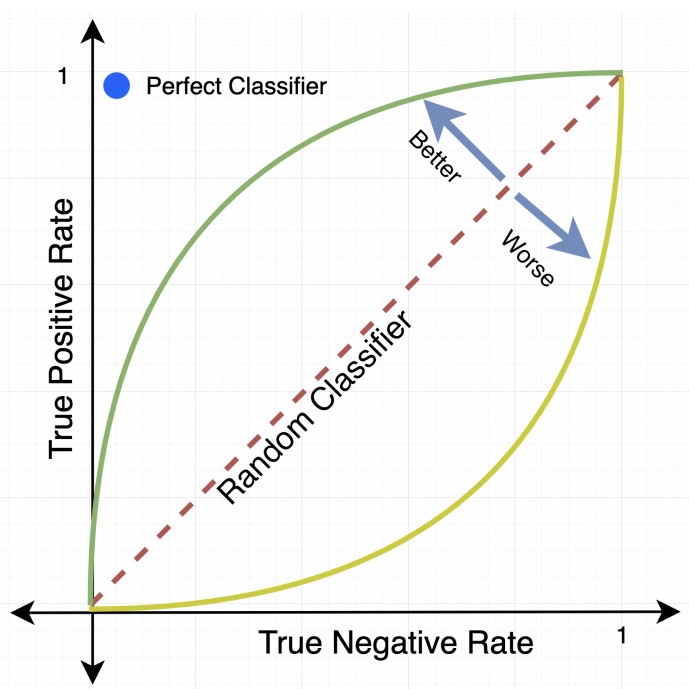

**Figure 24.** ROC curve interpretation.

*7.4. Accuracy vs. Additional Random Samples Testing*

    Since the cancer patients dataset shows the least-effective classifier, as shown in Figure 23, it was tested for its response to the injection of the a sudden spike in additional

random samples up to 100%. Figure 25 shows that, upon increasing the random samples, the result of PCA–SRP decreases gradually.

Figures 26 and 27 present the validation accuracy of both ANN + PCA(SRP) and ANN only to additional noises up to 100% with selectivities (Sc) of 88 and 98%, respectively. It has also shown a reasonable increase in accuracy using 98% selectivity (*Sc*) instead of 88%.

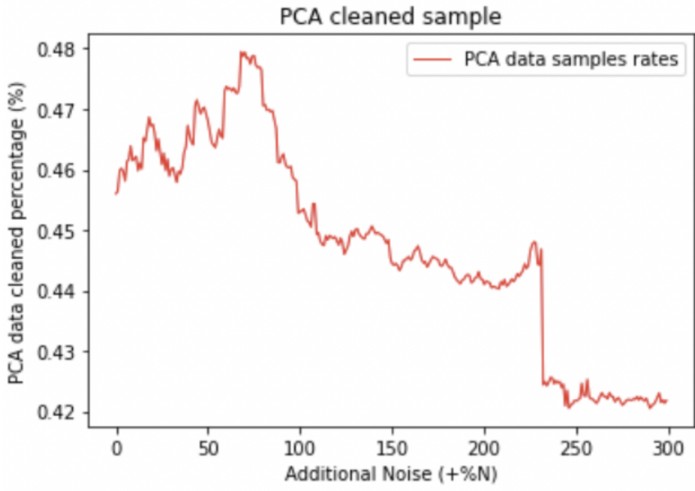

**Figure 25.** PCA cleaned data percentage (%).

The high-valued data have been preserved and maintain their accuracy until a specific additional noise point. Nevertheless, the predictive power remains intact until that point, even though some data were lost in the process.

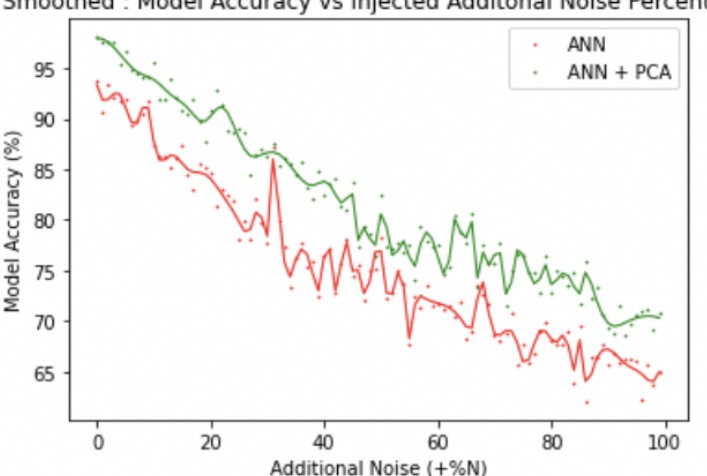

**Figure 26.** +%Noise response for cancer patients dataset (Sc = 88%).

The $ANN + PCA_{SRP}$ maintains the highest performance of cancer patient classifications with high *Sc* values. The methodology allows for a significant advantage by gradually lessening the decrease in classification problem accuracy over the sudden increase in noise in the system.

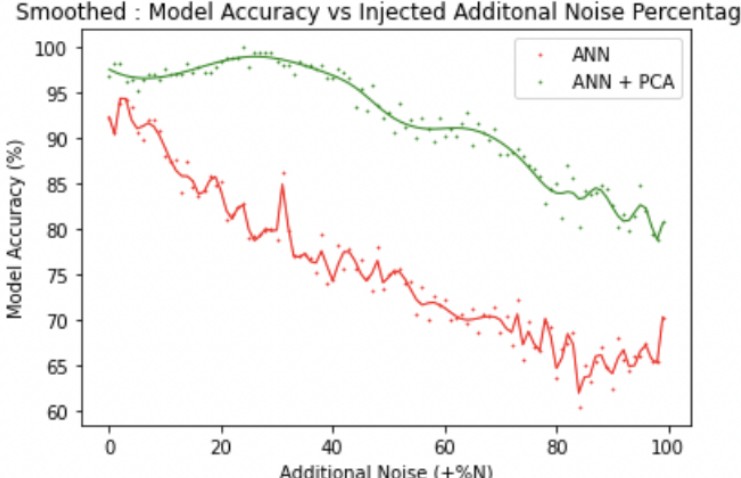

**Figure 27.** +%Noise response for the cancer patients dataset (Sc = 98%).

The ANN classification problem requires strong training sets, which requires a lot of high *Sc* while disregarding the low ones; based on the observation in both Figures 26 and 27, $Sc_{ANN}$ is better, as described in Equation (26).

## 8. Conclusions and Future Research

The material presented in this paper shows a significant improvement in the accuracy of an ANN in classification problems with the aid of principal component analysis–sample reduction process (PCA–SRP). The ANN cast off 10% of the learning rate, two layers with 32 and 16 hidden neurons, ReLU activators in hidden layers, and SoftMax in output activators with 100 epochs or iterations, as shown in Table 3 based on the PCA–SRP and ANN The Python implementation program was implemented on the multidimensional datasets gathered, namely the heart disease, gender voice recognition, breast cancer classification, and cancer patients datasets provided in Table 4. These datasets were then used in the PCA–SRP + ANN accuracy comparison testing, sensitivity vs. specificity testing, receiver operating characteristic (ROC) curve testing, and accuracy vs. additional random samples testing; the results show significant improvements.

PCA–SRP removed dirty and imprecise datasets, based on the results shown in Table 5, which allowed us to reduce the number of samples in the process and allowed for a significant increase in accuracy, as shown in Tables 6 and 7. Furthermore, we also determined the recommended *Sc* range values for normal cleaning and the ANN classification problem.

Future research will further investigate the performance in massive biomedical datasets and determine how to load them into PCA–SRP cleansing agents; one of the suggestions is loading through batch processing. Another suggestion is to utilise knowledge-based techniques of PCA–SRP in different neural network architectures such as CNNs, RNNs, LSTMs, and GNNs. Furthermore, an investigation into various field applications to explore the incorporation of the investigated cleaning techniques, such as real-time biomedical automation, image-based medical diagnosis classification, and human thoughts processes [48–51] could be a desirable research avenue.

The proposed methodology can be applied to a wearable EEG or similar device in order to extract chaotic data from the brain's unique biometric featured samples for use in cryptography or especially for steganography. Lastly, this formal basis is necessary to design and construct high-quality and helpful software tools to support the data cleansing process of PCA–SRP and its application for artificial neural networks (ANNs).

**Author Contributions:** Conceptualisation, C.M.S.A., T.Y.W. and H.C.; methodology, C.M.S.A., T.Y.W. and H.C.; software, C.M.S.A.; validation, H.C., T.Y.W. and S.A.-M.; formal analysis, C.M.S.A. and H.C.; investigation, C.M.S.A., T.Y.W. and H.C.; resources, H.C., T.Y.W. and S.A.-M.; data curation, C.M.S.A.; writing—original draft preparation, C.M.S.A.; writing—review and editing, C.M.S.A.,

T.Y.W. and H.C.; visualisation, C.M.S.A., T.Y.W. and H.C.; supervision, T.Y.W. and H.C.; project administration, C.M.S.A. and H.C.; funding acquisition, C.M.S.A. and H.C. All authors have read and agreed to the published version of the manuscript.

**Funding:** The publication of this article was funded by the Department of Science and Technology—Science Education Institute (DOST-SEI), Republic of the Philippines, through the Foreign Graduate Scholarship.

**Institutional Review Board Statement:** Not applicable.

**Informed Consent Statement:** The sample dataset is publicly available, but specific invisible health data will not be provided as consent was not given.

**Data Availability Statement:** The data presented in this study are available from the corresponding author upon request.

**Acknowledgments:** The authors thank Christofer Yalung for their encouragement of the first author to perform research and for their proof reading of the English spelling and grammar in our research paper. Furthermore, we thank the faculty members of the University of Gloucestershire who supported the authors in their research.

**Conflicts of Interest:** The authors declare no conflicts of interest.

**Sample Availability:** The sample datasets are available from the given URL in the references.

## Abbreviations

The following abbreviations are used in this manuscript:

| | |
|---|---|
| AI | Artificial Intelligence |
| ANN | Artificial Neural Network |
| AUC | Area Under Curve |
| CNN | Convolutional Neural Network |
| GNN | Graph Neural Network |
| LSTM | Long Short-Term Memory |
| PCA | Principal Component Analysis |
| ReLU | Rectified Linear Unit |
| RNN | Recurrent Neural Network |
| ROC | Receiver Operating Characteristic |
| SCARE | Scalable Automatic Repairing |
| SRP | Sample Reduction Process |

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
