# Peer review of "Sample Reduction for Physiological Data Analysis Using Principal Component Analysis in Artificial Neural Network"

_applsci, doi:10.3390/app11178240_

Round 1
Reviewer 1 Report
The find this paper an interesting read. Applying PCA for reduction or cleaning of noisy data is intuitive. However, I have the following comments:
- My major concern is regarding the scalability of the approach. The authors have shown improvement in accuracy after applying the method however all the datasets examined are small. How can this be applied to larger datasets when it is difficult to load all samples at once?
- If there are too many noisy samples in a small dataset the approach will not help much. How do the authors propose the method be used in such case? Please address this in the manuscript.
- The write-up lacks clarity in some sections. e.g. line 247 does not make much sense.
Author Response
Thanks for the very wonderful review of my paper for very good insight and suggestions. If there are anything more sections that need revision please don't hesitate to inform me. I am glad to revise it to the best of my writing ability.
Herewith attached is the point by point response to the reviewer.

Reviewer 2 Report
Title: Sample reduction for physiological data analysis using principal component analysis in ANN
Summary:
Authors proposed a sample reduction process using PCA to reduce the noise percentage and hence improve the predictive accuracy of ANN for physiological data analysis. The proposed technique is demonstrated on clinical trial data sets. The outcome shows improvement in noise reduction and prediction capability.
Comments:
- In section 6, PCA is always mentioned rather than PCA-SRP. Reviewer assumed the authors meant to use PCA-SRP rather than PCA in the outcomes presented in Section 6. Have the authors tried PCA without SRP and compare with PCA with SRP? It will be good to compare the three outcomes from ANN, without PCA, PCA without SRP and PCA with SRP for a better discussion on the strength or capability of PCA with SRP.
- Reviewer strongly suggest the authors to proof read and have extensive editing on the English for the manuscripts before next submission. Avoid using ‘We believe’, ‘can’t’, etc. as they don’t sound professional in a journal paper. Be consistent with your notation and wordings, “multidimensional” or “multi-dimensional”? “SC” or “Sc”? “*” or “×”? “RELU” or “ReLU”
- Most of the figures are in very low resolution, low visibility and low readability. Can the authors improve the quality of the presented figures?
- Start “F”igures and “T”ables with Cap in the text.
- Thought on the results in Fig. 18 and Fig. 26 (ANN+PCA). Did the authors overfit the data?
- Apologise if the reviewer didn’t find the context in the manuscript. The authors mentioned about setting threshold to accept or reject the data but the reviewer cannot find any description on how these thresholds were set. Is the threshold a static or dynamic threshold? Does the threshold change with different sample or just manually set up? How will the threshold affect the accuracy? How did the authors justified about the threshold setting too high or too low? Are they related to Sc in Section 5?
- The captions for each Figure have to be reviewed again. Fig 1 and Fig 2 are having the same captions! And both of them did not fully describe what has been presented.
- The sequence for the references is all messed up!
- Not all the notations and abbreviations are addressed properly in the manuscripts and since abbreviations are introduced, the manuscript should use them wisely!
- What is “top” PCA in Eqn. 3?
- Fig 3, which direction is X and Y?
Author Response
Thanks for the very wonderful review of my paper writing, and very good insight and suggestions. If there are anything more sections that need revision please don't hesitate to inform me. I am glad to revise it to the best of my writing ability.
Herewith attached is the point by point response to the reviewer.

Round 2
Reviewer 1 Report
The authors have addressed my comments in the rebuttal. It would have been better if authors had results on a larger dataset however I appreciate addressing this in the paper.
I have one suggestion to include reference to other experimental areas which have smaller datasets and this technique can be used for dataset cleaning. This can improve the visibility and application options for the study. Some suggestions include: a) Vyas, Shruti, Subhabrata Das, and Yen-Peng Ting. "Predictive modeling and response analysis of spent catalyst bioleaching using artificial neural network." Bioresource Technology Reports 9 (2020): 100389. b) Tirupattur, Praveen, et al. "Thoughtviz: Visualizing human thoughts using generative adversarial network." Proceedings of the 26th ACM international conference on Multimedia. 2018.
Author Response
Thanks for your comments, insights, and suggestion. I truly appreciate it.

Reviewer 2 Report
The authors have addressed the comments accordingly.
Author Response
Thank you again, for your wonderful insight, comments, and suggestion. I truly appreciate it.